# *SLC35A2*-Related Brain Disorders: Genetics, Pathophysiology, and Therapeutic Insights

**DOI:** 10.3390/ijms262311560

**Published:** 2025-11-28

**Authors:** Beatrice Risso, Antonella Riva, Greta Volpedo, Valerio Conti, Clara Tuccari di San Carlo, Federico Zara, Pasquale Striano, Antonio Falace

**Affiliations:** 1Medical Genetics Unit, IRCCS Istituto Giannina Gaslini, Full Member of ERN-EPICARE, 16147 Genoa, Italy; 2Department of Neurosciences, Rehabilitation, Ophthalmology, Genetics, Maternal and Child Health, University of Genoa, 16132 Genoa, Italy; 3Department of Neuroscience and Human Genetics, Meyer Children’s Hospital IRCCS, Viale Pieraccini 24, 50139 Florence, Italy; valerio.conti@meyer.it (V.C.); clara.tuccari@meyer.it (C.T.d.S.C.); 4Pediatric Neurology and Muscular Diseases Unit, IRCCS G. Gaslini, Full Member of ERN-EPICARE, 16147 Genoa, Italy

**Keywords:** *SLC35A2*, MOGHE, epilepsy

## Abstract

*SLC35A2* encodes the Golgi uridine diphosphate galactose transporter, which is essential for glycosylation of glycoproteins and glycolipids. Variants in this gene, either germline or somatic, have emerged as causes of diverse neurological disorders ranging from congenital disorders of glycosylation (*SLC35A2*-CDG) to focal cortical malformations such as mild malformation of cortical development with oligodendroglial hyperplasia in epilepsy (MOGHE). This review summarizes the molecular function of *SLC35A2*, clinical phenotypes of congenital and somatic variants, insights from functional assays and animal models, and therapeutic perspectives including galactose supplementation and precision medicine. We aim to provide an integrative synthesis of human genetics, neuropathology, glycomics, and translational approaches.

## 1. Introduction: Protein Glycosylation in the Brain

Glycosylation represents one of the most abundant and functionally diverse post-translational modifications in the central nervous system (CNS). Two major types of protein glycosylation occur in the brain: N-glycosylation, where glycans are attached to the asparagine residue of a consensus sequence (Asn-X-Ser/Thr), and O-glycosylation, in which glycans are linked to serine or threonine residues [1,2,3]. Both types of glycan moieties are synthesized and modified by glycosyltransferases located in the lumen of the endoplasmic reticulum (ER) and Golgi network [1,2]. While N-glycosylation is essential for protein folding, stability, and trafficking, O-glycosylation regulates cell–cell interactions, receptor signaling, and synaptic plasticity [3,4]. In the CNS, glycoproteins are enriched at synapses, in myelin, and at the neuronal cell surface, where they influence axon guidance, neurotransmission, and myelination [1,5]. Altered N-glycosylation can impair folding and trafficking of ion channels and neurotransmitter receptors, contributing to epilepsy and neurodevelopmental delay [6]. O-glycosylation defects, particularly in mucin-type O-glycans and O-mannosylation, have been linked to muscular dystrophies with brain involvement, cortical malformations, and abnormal synaptic function [7]. The brain, with its high metabolic demands and complex cellular architecture, is particularly sensitive to glycosylation defects [8]. Thus, precise regulation of glycosylation pathways is essential for brain physiology, and disruption leads to a wide spectrum of neuropathological phenotypes collectively referred to as congenital disorders of glycosylation (CDG) [9,10,11].

## 2. The SLC35 Protein Family and SLC35A2

The SLC35 family belong to the solute carrier (SLC) superfamily, constituted by transmembrane proteins involved in the translocation of a wide range of substrates across biological membranes [12,13]. In humans, the SLC35 family is divided into seven subfamilies (SLC35A-G), mainly represented by nucleotide sugar transporters (NSTs) [12,13]. At ER-Golgi glycosyltransferases utilize nucleotide sugars such as uridine guanosine diphosphate fucose (GDP-Fuc), uridine diphosphate galactose (UDP-Gal), UDP-glucose (UDP-Glc) and UDP-N-acetylglucosamine (UDP-GlcNAc) as glycosylation substrates [14,15]. These nucleotides, however, are synthesised in the cytosol and they have to be transported into the ER-Golgi trough NSTs [16,17,18]. Among the SLC35 family, the SLC35A subfamily is considered to be exclusive composed by NSTs [13]. The genes of this subfamily encode for CMP-sialic acid transporter (CST, SLC35A1), UDP-galactose and UDP-N-acetylgalactose transporter (UGT, SLC35A2), UDP-N-acetylglucosamine transporter (NGT, SLC35A3), and CDP-ribitol transporter (SLC35A4) [13,19]. No substrate specificity has been discovered yet for SLC35A5 [19]. However, several aspects of SLC35 protein family biology remain incompletely understood. For instance, the structural determinants of NST oligomerization, the mechanisms regulating the formation and stability of heterologous transporter complexes, and the extent to which Golgi metabolic status modulates transporter activity still need to be defined. Addressing these open questions will be essential to fully elucidate how NSTs coordinate glycosylation within the secretory pathway. The *SLC35A2* gene, located on chromosome Xp11.23, encodes the UGT protein. This gene produces two isoforms, UGT1 and UGT2, which arise through alternative splicing and differ in their C-terminal sequences. UGT1 is localized exclusively to the Golgi apparatus, whereas UGT2 is distributed between both the Golgi and the ER [20]. Structurally, SLC35A2 is a multi-pass transmembrane protein predicted to contain 10 transmembrane helices, organized into two pseudo-symmetrical halves [13,21]. Like other NSTs, it likely functions as an oligomer, with evidence suggesting that dimerization or higher-order assembly is required for optimal transport activity and Golgi localization [22,23,24] (Figure 1). Interactions between SLC35A2 and other Golgi-resident NSTs may also contribute to substrate specificity and functional regulation [24]. Pathogenic variants in *SLC35A2*, whether germline or somatic, have been increasingly recognized as causes of a spectrum of neurological disorders (Figure 2 and Appendix A). These range from systemic conditions such as congenital disorders of glycosylation (*SLC35A2*-CDG) to focal cortical malformations, most notably mild malformation of cortical development with oligodendroglial hyperplasia in epilepsy (MOGHE). Germline and somatic variants do not produce sharply distinct syndromes, but somatic mutations restricted to focal brain regions explain the highly localized seizure onset seen in some patients [2,3,11]. Broadly, *SLC35A2* variants fall into two categories: (i) germline or de novo mutations, which give rise to *SLC35A2*-CDG and result in widespread neurodevelopmental impairment, and (ii) somatic, brain-restricted mutations, which underlie focal cortical malformations such as MOGHE and non-lesional focal epilepsies. This duality highlights different pathogenic mechanisms: while germline variants impair glycosylation in all tissues and result in multisystemic disease, somatic mosaicism produces localized defects in neuronal and glial lineages, emphasizing its role as a key driver of focal epilepsies.

In this review, we will: (1) provide an overview of congenital and somatic *SLC35A2* variants; (2) examine the role of somatic mosaicism in MOGHE; (3) summarize functional studies in patient tissues and variant assays; (4) review insights from animal models; and (5) discuss therapeutic perspectives, including metabolic supplementation and precision medicine approaches.

## 3. From CDG to MOGHE: The Spectrum of SLC35A2-Related Disorders

### 3.1. SLC35A2 Germline and Systemic Mosaic Variants

Congenital variants in *SLC35A2* were first reported by Kodera et al. [25], who identified de novo mutations in females with early-onset epileptic encephalopathy (EOEE) [25]. Subsequently, Ng et al. identified systemic post-zygotic mosaic variants in both males and females, highlighting that pathogenic *SLC35A2* variants can also occur as systemic mosaicisms, which may influence disease severity and viability [26]. Since that initial description, additional cohorts have confirmed a consistent clinical picture characterized by severe epilepsy and global developmental impairment [27,28,29,30,31]. Seizures usually begin in the neonatal period or early infancy and may include infantile spasms, focal seizures, or generalized seizures, frequently progressing to drug-resistant epilepsy. Electroencephalographic findings such as hypsarrhythmia and multifocal epileptiform activity are common [27]. A striking female predominance has been noted, consistent with the X-linked inheritance pattern and reduced viability of males carrying hemizygous germline mutations [27,28,29,30,31]. Biochemical analyses in congenital cases often demonstrate defective glycosylation, particularly reduced galactosylation of N-glycans, although routine transferrin testing can be normal in some individuals, making genetic sequencing essential for diagnosis [27,28]. Functional studies have provided direct mechanistic evidence: complementation assays, glycan profiling, and rescue experiments consistently show that both missense and truncating variants impair UDP-galactose transport, leading to hypogalactosylation of glycoproteins [28]. Larger series have also revealed that residual transporter activity may modulate disease severity, although the epileptic phenotype remains a hallmark across variant types [26,27]. Therapeutically, galactose supplementation has been trialed in SLC35A2-CDG, with reports of improved biochemical profiles and, in some patients, reduced seizure frequency and developmental gains, though these preliminary results require validation in controlled settings [32].

### 3.2. Somatic SLC35A2 Mosaicism in the Brain

Somatic, brain-restricted variants in *SLC35A2* were first identified by Winawer et al., who reported mosaic pathogenic mutations confined to resected cortical tissue from patients with pharmacoresistant focal epilepsy, absent in blood-derived DNA, thus establishing the concept of brain-only *SLC35A2* mosaicism [33] (Figure 3 and Figure 4). Shortly thereafter, Sim et al. corroborated these findings and provided biochemical evidence that affected cortical tissue exhibited defective N-glycan galactosylation, directly linking somatic SLC35A2 dysfunction to aberrant glycosylation in the epileptogenic cortex [34]. Building on these observations, Baldassari et al. applied ultra-deep sequencing of paired brain and blood samples in surgical cases of mild Malformation of Cortical Development/Focal Cortical Dysplasia Type I (mMCD/FCD1) and Focal Cortical Dysplasia Type II/Hemimegaencephaly (FCD2/HME) [35]. They showed that nearly one-third of mMCD/FCD1 lesions of the cohort carried somatic loss-of-function *SLC35A2* variants, absent from blood and without evidence of mTOR pathway hyperactivation, while FCD2/HME cases were consistently associated with activating mTOR mutations and pS6 hyperactivation. These findings established *SLC35A2*-related mMCD/FCD1 as a mechanistically distinct subgroup of cortical malformations, independent of mTOR signaling and separate from the canonical FCD2/HME spectrum.

Subsequent studies have expanded the genetic and clinical landscape of brain *SLC35A2* mosaicism. Variants are typically deleterious, including nonsense, frameshift, splice-site, or missense substitutions, detected at low to moderate variant allele fractions in resected cortex, and generally undetectable in peripheral blood [36,37,38]. Large multicenter surgical series have refined the clinical and electrophysiological correlates. In particular, Barba et al. reported a multicenter cohort of 47 patients with brain somatic *SLC35A2* variants and featuring complex epileptic phenotype ranging from early epileptic encephalopathy (EE, 39 patients) with epileptic spasms to drug-resistant focal epilepsy (DR-FE, 8 patients) associated with normal/borderline cognitive function [38]. The 47 patients, which represents the most comprehensive datasets available, harbored 42 distinct brain mosaic *SLC35A2* variants, including 14 (33.3%) missense, 13 (30.9%) frameshift, 10 (23.8%) nonsense, 4 (9.5%) in-frame deletions/duplications, and 1 (2.4%) splicing variant. Variant allele frequencies (VAFs) ranged from 1.4% to 52.6% (mean: 17.3 ± 13.5).

Electroencephalography (EEG) in these cohorts typically mirror seizure semiology showing focal epileptiform discharges concordant with the epileptogenic zone with occasional secondary generalization and variably slowed background activity. Conversely, MRI findings were variable, ranging from normal appearances on routine evaluation to subtle cortical abnormalities, such as mild cortical thickening, blurring of the grey–white matter junction, or signal changes suggestive of cortical developmental malformations. Notably, complete seizure freedom was achieved in 61.5% of patients with EE and in 37.5% of those with DR-FE after surgical resection. Their data reinforced the clinical relevance of recognition of early somatic variants in *SLC35A2*, emphasizing the translational significance of integrating molecular genetics with detailed tissue analysis. Importantly, seizure freedom was not correlated with VAF levels or with variant class, underscoring that even low-level mosaicism can sustain epileptogenic networks and still benefit from surgical resection. Post-operative cognitive status remained largely stable, confirming surgery as an effective strategy for seizure control rather than neurodevelopmental improvement [38].

Further pathological and experimental work has confirmed the mechanistic basis of *SLC35A2* physiopathology. Resected cortical samples with *SLC35A2* mutations exhibit altered glycosylation patterns and abnormal cortical cytoarchitecture [36], supporting a model in which impaired UDP-galactose transport during brain development disrupts neuronal networks and establishes a focal epileptogenic substrate. The study of Miller et al. highlighted region-specific differences, with the hippocampus and parietal/occipital lobes showing variable involvement on neuroimaging, EEG, and histopathology, linking focal biochemical defects to epileptogenic zones [36].

More recent reviews and series have strengthened the evidence that somatic *SLC35A2* mutations are not only recurrent but among the most frequent genetic alterations identified in otherwise non-lesional or subtly lesional focal epilepsies undergoing surgical evaluation [35,39]. Collectively, in vivo models indicate that loss of SLC35A2 function affects multiple neural cell types, including both neurons and oligodendroglial lineages. Current evidence suggests a prominent involvement of excitatory neurons, although contributions from oligodendroglial alterations to network instability and epileptogenesis cannot be excluded and warrant further investigation. These complementary findings delineate a neuron-centric mechanism underlying cortical dyslamination and network hyperexcitability in *SLC35A2*-related malformations.

### 3.3. SLC35A2-Associated MOGHE

MOGHE was first formally described by Schurr et al., as a distinct clinico-pathological entity [40]. The study detailed children and young adults with frontal lobe epilepsy whose resected tissue showed oligodendroglial hyperplasia (Olig2+), clusters of heterotopic neurons in the white matter, and blurred grey–white matter boundaries, without dysmorphic neurons or balloon cells characteristic of FCD type II (Figure 5). This work distinguished MOGHE from classical FCD subtypes and established it as a separate entity. Blümcke and colleagues subsequently contributed to its formal recognition within the revised ILAE classification of FCD and lesional epilepsies [41], defining MOGHE as a variant with mild cortical malformation and oligodendroglial hyperplasia, distinct from both FCD types I and II.

Early neuroradiological studies, such as those by Hartlieb et al., described ill-defined T2/FLAIR hyperintensities at the grey–white matter junction, often age-dependent, with infantile-onset cases presenting with epileptic spasms and older children exhibiting focal seizures [42]. Recent studies and reviews have further contextualized MOGHE within the spectrum of cortical malformations. Gaballa et al. reported the first systematic clinico-surgical series of 20 patients with histologically confirmed MOGHE, further refining its clinical and epileptological classification [37].

In their cohort seizures typically began in infancy, though some presented later, and were characterized by early spasms followed by focal hyperkinetic seizures. Interictal EEG often showed frequent unilateral slow spike or polyspike activity during sleep and 2–2.5 Hz spike-wave paroxysms during wakefulness. Lesions were mostly frontal and sometimes initially undetectable on standard imaging. Postoperative seizure freedom was achieved in ~60% of patients, with better outcomes associated with earlier surgery and more extensive resection [37]. This study highlighted that clinical, EEG, and imaging features, alongside histology, can support the preoperative recognition of MOGHE, reinforcing its status as a distinct entity and guiding surgical management.

The genetic basis of MOGHE was first elucidated by Bonduelle et al., who analyzed resected cortical tissue from pediatric patients with pharmacoresistant focal epilepsy and histopathological MOGHE [43]. They identified pathogenic somatic *SLC35A2* variants in in 9/20 (45%) patients with VAFs ranging from very low (<5%) to higher levels (>20%). Variants were absent from blood-derived DNA, confirming their post-zygotic, somatic origin. Functional validation using lectin histochemistry demonstrated focal hypogalactosylation in neurons and glial cells, providing a molecular correlate to the observed histopathological abnormalities [43]. Genetic evidence thus transformed what had been an ambiguous category into a coherent and distinct entity within the spectrum of lesional epilepsies.

These findings have driven a broad histopathological reassessment. Cases initially considered borderline or atypical FCD have been reclassified as MOGHE, finally integrating morphological, clinical, and molecular data into a coherent diagnostic framework consistent with the revision of the ILAE classification of FCD and lesional epilepsies [41]. The clinical relevance of this reclassification is highlighted in the *SLC35A2* cohort reported by Barba et al., 2023 where histopathology review in the cohort identified MOGHE in 44/47 patients [38]. In their meta-analysis of 163 histologically confirmed MOGHE cases from 18 studies, Zhan et al. provided the largest synthesis to date, defining the core clinical and molecular features of the disorder [44]. The median age at seizure onset was 1.2 years, with a slight male predominance and predominantly unilobar, frontal lesions. Approximately 73% of cases with available molecular data harbored somatic *SLC35A2* variants restricted to brain tissue, reinforcing the causal role of mosaic dysfunction in UDP-galactose transport. *SLC35A2*-positive patients more frequently presented with infantile epileptic spasms and Lennox–Gastaut syndrome phenotypes, while later-onset cases tended to show focal seizures. Postoperative seizure freedom or marked improvement was achieved in about 60% of cases, confirming surgery as an effective therapeutic approach [44].

These results consolidate MOGHE as a distinct clinico-pathological and molecular entity within the spectrum of *SLC35A2*-related cortical malformations, characterized by early-onset epilepsy and favorable surgical outcomes. While somatic *SLC35A2* variants account for the majority of genetically characterized cases, a proportion of histologically confirmed MOGHEs lack detectable mutations even with high-sensitivity sequencing, indicating genetic heterogeneity. This suggests that additional pathogenic mechanisms—potentially involving other genes regulating glycosylation, cortical development, or oligodendroglial proliferation, as well as epigenetic or regulatory alterations—may converge on a shared histopathological and clinical phenotype.

Remarkably, recent preprint findings further expand the genetic landscape of MOGHE by demonstrating that, in 29 brain tissue samples from individuals with histopathologically confirmed MOGHE, clusters of oligodendroglial hyperplasia exhibited significant Y-chromosome gains in approximately 84% of males and 50% of females, whereas the overlying non-lesional cortex remained unaffected [45]. These observations raise the possibility that sex-chromosome-linked mechanisms might contribute to MOGHE, but further replication and functional studies are needed before drawing mechanistic conclusions. Combining these Y-chromosome gains with pathogenic *SLC35A2* variants allowed stratification of patients into three genetically defined subgroups, which showed differences in age at seizure onset and lesion volume, highlighting a previously underappreciated contribution of sex chromosome biology to MOGHE pathogenesis.

Collectively, these insights support defining MOGHE primarily by its clinico-pathological signature, while emphasizing the need for integrative multi-omic approaches to uncover the full molecular spectrum of this distinct epileptogenic disorder.

Taken together, current data establish *SLC35A2* variants as major contributors to both systemic CDG and focal cortical malformations, yet important uncertainties persist. These include the true prevalence of brain mosaicism, the impact of sampling limitations on variant detection, and the molecular bases of *SLC35A2*-negative MOGHE. Furthermore, the relative contribution of neuronal and oligodendroglial dysfunction to epileptogenesis remains an open question. Future studies integrating deep sequencing, spatial transcriptomics, and standardized neuropathological criteria will be crucial to refine disease classification

## 4. Molecular and Cellular Pathogenesis of SLC35A2-Related Disorders

### 4.1. Insights from Human and Induced Pluripotent Stem Cell (iPSC) Models of MOGHE

Recent work has provided striking insights into the molecular and cellular consequences of somatic *SLC35A2* variants in human brain tissue, bridging the gap between altered glycosylation, cell-type-specific dysfunction, and epileptogenic network remodeling. Collectively, these studies delineate a coherent pathogenic continuum that integrates biochemical, cellular, and transcriptional data from both patient specimens and human neuronal models.

Using complementary approaches, recent investigations have progressively clarified how SLC35A2 dysfunction drives cortical pathology in MOGHE. Cecchini et al. mapped the spatial distribution of the SLC35A2 protein in surgical samples, confirming its predominant Golgi localization across neurons and glia and showing that different variant classes lead to distinct expression patterns: nonsense mutations cause marked loss of signal, while missense variants retain the protein but with abnormal perinuclear accumulation, consistent with defective intracellular trafficking [46]. Liu et al. extended these findings by coupling genetic screening with glycoproteomic and histochemical analyses, demonstrating that all identified somatic variants result in a consistent shift toward truncated, agalactosylated N-glycans and focal depletion of complex galactosylated structures, particularly within heterotopic neurons [47]. The affected glycoproteins were enriched in pathways related to cell adhesion, axon guidance, and myelination, linking impaired glycosylation to the cellular disorganization and white matter changes typical of MOGHE.

Lai et al. further confirmed these pathogenic effects in isogenic iPSC derived neurons carrying *SLC35A2* variants, where loss of UDP–galactose transport led to reduced N-glycan branching, premature neuronal differentiation, and hypoactive, poorly synchronized network activity [48].

A complementary perspective was provided by Galvão et al., who applied single-nucleus RNA and Assay for Transposase-Accessible Chromatin using sequencing (ATAC-seq) to MOGHE lesions [49]. Although the analyzed samples did not harbor detectable somatic *SLC35A2* variants, their multimodal profiling revealed convergent transcriptional and epigenetic alterations within oligodendrocyte-lineage cells and heterotopic neurons. Oligodendrocyte clusters exhibited reduced expression of myelin-related genes and activation of stress-response pathways, while displaced neuronal populations showed dysregulation of adhesion and cytoskeletal programs. These changes delineate the cellular compartments and molecular processes most perturbed in MOGHE and, despite the absence of confirmed *SLC35A2* mutations, point to shared downstream mechanisms involving disturbed glycosylation-dependent functions and cell-type-specific stress adaptations. In this way, the findings of Galvão et al. extend the pathogenic model of *SLC35A2*-related MOGHE, suggesting that similar patterns of molecular reprogramming and network vulnerability may arise through partially overlapping pathways within the epileptogenic cortex.

Taken together, these complementary datasets outline a unified pathogenic cascade: somatic *SLC35A2* variants lead to reduced or mislocalised Golgi transporter protein, impairing UDP–galactose import and thereby truncating N-glycan synthesis on key membrane and secreted glycoproteins. The resulting hypogalactosylation alters adhesion, migration, and myelination processes, which manifest as oligodendroglial hyperplasia, neuronal heterotopia, and transcriptional remodeling in affected cortical regions. Ultimately, these molecular and cellular perturbations converge at the network level, producing an asynchronous, dysregulated microcircuit that provides the substrate for epileptogenesis. This integrated evidence positions defective glycosylation as the central mechanistic link between mosaic *SLC35A2* mutations and the structural–functional continuum of MOGHE.

### 4.2. In Vivo Mechanistic Studies of SLC35A2 Pathology

The mechanistic framework derived from human and iPSC-based studies has been substantially expanded through a series of in vivo models exploring how SLC35A2 dysfunction affects cortical development and network organization. Approaches based on in utero electroporation (IUE) or conditional gene ablation, mutations have collectively provided consistent evidence that impaired SLC35A2 activity disrupts cortical architecture and alters neuronal excitability.

Two independent studies (Elziny and Spyrou) using IUE-based *Slc35a2* loss-of-function models showed consistent laminar disorganization and dendritic abnormalities, without overt seizures but with localized epileptiform activity [50,51]. Despite these cytoarchitectural abnormalities, no spontaneous seizures were detected in these mouse models, suggesting that focal *Slc35a2* deficiency primarily induces structural and microcircuit-level alterations without progressing to overt epilepsy [51]. However, chronic EEG monitoring in the *Slc35a2*-knockdown model as well as local field potential recordings detected intermittent epileptiform discharges confined to electroporated zones, while ex vivo patch-clamp analyses revealed altered intrinsic excitability of *Slc35a2*-deficient neurons and reduced local network synchrony [51]. Together, these findings define a state of latent hyperexcitability that may precede overt seizure activity and mirror the network desynchronization described in *SLC35A2*-mutant human iPSC-derived neurons, suggesting convergent mechanisms of impaired excitability control.

Further insights into the cell-type-specific contributions of SLC35A2 have come from conditional knockout approaches in mice. Yoon et al. generated *Slc35a2* conditional knockouts targeting either excitatory neuron progenitors (*Emx1*-Cre) or oligodendrocyte-lineage cells (*Olig2*-Cre) to dissect the respective roles of neuronal and glial dysfunction [52]. Neuron-specific deletion resulted in laminar disorganization, migration defects, and spontaneous seizures, whereas *Olig2*-driven deletion caused only mild myelin abnormalities without overt epilepsy. Independently, Bartel et al. re-examined SLC35A2 function in oligodendrocytes using a similar *Slc35a2^Olig2^*^-Cre^ mouse model, but aimed at defining the role of SLC35A2 in oligodendrocyte development and myelin maintenance [53]. Their study revealed diffuse hypomyelination, reduced oligodendrocyte density, and spontaneous seizures associated with interictal EEG abnormalities [53]. Although both groups used an *Olig2*-Cre strategy, differences in genetic background, recombination timing, and experimental endpoints likely contributed to the distinct phenotypes. Rather than conflicting, these findings highlight complementary aspects of SLC35A2 biology, indicating that both neuronal and oligodendroglial dysfunction can impair network organization and excitability through partially overlapping mechanisms. Together, these studies delineate a broader pathogenic framework in which defective glycosylation disrupts cortical architecture and compromises network stability across multiple neural cell types.

Finally, Falace et al. established a rat model of *SLC35A2*-related cortical malformation by introducing patient-derived missense variants through IUE [54]. This strategy enabled the creation of focal cortical mosaics in which *SLC35A2*-mutant neurons coexist with wild-type neighbouring cells, thereby mimicking the somatic distribution pattern observed in patients. As observed in mice models, in vivo acute *Slc35a2* knockdown in rat neuroprogenitors impaired neuronal migration. In parallel, neurons electoporated with human *SLC35A2* pathogenic variant displayed abnormal positioning, altered dendritic architecture, and disrupted local cytoarchitecture, collectively indicating that disease-associated variants can reproduce key developmental abnormalities characteristic of *SLC35A2*-related malformations [54].

Taken together, these animal studies converge on a hierarchical model in which SLC35A2 dysfunction perturbs neuronal differentiation and circuit integration, leading to localized hyperexcitability and, in specific contexts, spontaneous seizures. While the precise biochemical consequences on glycosylation remain to be directly demonstrated in vivo, the cross-consistency between animal and human data underscores the pivotal role of SLC35A2 in maintaining cortical organization and excitatory–inhibitory equilibrium.

## 5. D-Galactose Supplementation as a Targeted Treatment

Oral D-galactose supplementation currently represents the only disease-specific therapeutic approach tested in *SLC35A2*-related disorders, beyond conventional antiepileptic drugs and surgical interventions. The strategy is based on the ability of exogenous galactose to increase intracellular UDP-galactose levels through the Leloir pathway [55], thereby enhancing substrate availability for residual UDP-galactose transport into the Golgi and partially restoring glycosylation efficiency.

The first clinical evidence supporting this approach came from Witters et al., who evaluated oral galactose supplementation in patients with congenital *SLC35A2*-CDG [29]. Ten individuals received escalating doses up to 1.5 g/kg/day for 18 weeks. Treatment led to significant improvements in the Nijmegen Pediatric CDG Rating Scale, particularly in developmental and neurological domains, with several patients showing regained motor and feeding abilities. Biochemically, therapy increased the proportion of fully galactosylated N-glycans and reduced under-galactosylated structures [29]. This study provided the first proof-of-principle that galactose can bypass the metabolic bottleneck caused by impaired UDP-galactose transport, improving both biochemical and clinical parameters in congenital forms of the disease.

Building upon this rationale, Aledo-Serrano et al. tested D-galactose supplementation in patients with MOGHE who continued to experience seizures or cognitive deficits after epilepsy surgery [56]. Twelve individuals were treated for six months (up to 1.5 g/kg/day); six carried somatic *SLC35A2* variants in resected brain tissue, while six were genetically negative. Treatment was well tolerated, and all *SLC35A2*-positive patients (6/6) showed clinical improvement, including ≥50% seizure reduction and better cognitive or behavioral performance, with EEG improvement in half of the cases [56]. Interestingly, approximately half of the *SLC35A2*-negative patients also responded, suggesting that undetected low-level mosaicism or defects in other glycosylation-related genes might contribute to the same pathophysiological cascade [56]. These findings extend the potential of galactose therapy beyond congenital systemic forms to somatic, brain-restricted epilepsies, supporting its use as an adjuvant metabolic therapy after unsuccessful surgery.

Further mechanistic insights were provided by Pedrayes et al., who analyzed fibroblasts from patients with *SLC35A2*-CDG to elucidate the molecular basis of galactose rescue [57]. While protein N- and O-glycosylation defects were modest, cells exhibited severe abnormalities in glycosphingolipid and ganglioside biosynthesis, including accumulation of glucosylceramide and depletion of complex galactosylated species. Galactose supplementation normalized the lipid glycosylation profile by replenishing intracellular UDP-galactose and restoring glycosphingolipid synthesis, an effect mirrored by abnormal ganglioside patterns detected in patient serum [57]. Given the critical role of gangliosides in neuronal membrane organization and synaptic signaling, these data suggest that neurological improvement under galactose therapy may derive, at least in part, from the correction of ganglioside metabolism in the brain.

Collectively, these studies demonstrate that oral D-galactose supplementation can ameliorate both biochemical and clinical abnormalities across the spectrum of *SLC35A2*-related disorders, from congenital multisystemic forms to somatic epileptic malformations. Although evidence remains preliminary and derived from small, uncontrolled cohorts, the consistency of clinical benefit and favorable safety profile justify further exploration of galactose therapy as a targeted metabolic intervention for glycosylation-related epileptic encephalopathies.

Although D-galactose supplementation currently represents the only targeted intervention evaluated in patients, additional therapeutic strategies may merit consideration. Gene replacement approaches (e.g., adeno-associated virus-mediated *SLC35A2* delivery) or mRNA-based restoration of transporter expression could theoretically compensate for loss-of-function in mosaic cortical tissue, but remain at an early conceptual stage due to challenges in achieving efficient and cell-type-specific CNS delivery. Likewise, metabolic or small-molecule strategies aimed at enhancing UDP–galactose availability or stabilizing residual SLC35A2 activity may complement substrate supplementation.

## 6. Limitations and Additional Considerations

An important limitation inherent to brain-somatic variant studies is the potential sampling bias introduced by epilepsy surgery. Resected tissue represents only the clinically defined epileptogenic zone and may not fully capture the spatial distribution of mosaic variants across the lesion. Low-level somatic mutations may remain undetected if confined to adjacent or deeper cortical layers not included in the specimen or if preferentially present in specific cell populations under-sampled during routine neuropathology workflows. Therefore, absence of a detectable *SLC35A2* variant in resected tissue does not entirely exclude its presence elsewhere in the malformed cortex. Furthermore, technical variability among centers also contributes to differences in detection sensitivity. Sequencing depth, bioinformatic pipelines, DNA input quality, and the use of frozen versus FFPE tissue significantly influence the ability to identify low-variant-allele-fraction mosaicism. Ultra-deep sequencing or single-cell approaches are not uniformly available, which may lead to underestimation of mosaic *SLC35A2* variants in certain cohorts. Harmonization of analytical protocols and adoption of standardized thresholds for somatic variant calling will be essential to reduce inter-center variability and improve diagnostic precision.

A proportion of histologically confirmed MOGHE cases remain negative for *SLC35A2* variants even after high-depth sequencing. Several alternative molecular mechanisms may underlie these unresolved cases. Undetected ultra-low-level mosaicism may escape bulk sequencing approaches and require single-cell or spatial genomics for detection. Pathogenic variants may affect regulatory, promoter, or enhancer regions controlling *SLC35A2* or other glycosylation-related genes, which are not typically covered by clinical gene panels. Convergent defects in components of UDP–galactose metabolism, Golgi trafficking, or glycosyltransferase networks could theoretically produce a similar clinico-pathological phenotype. Preliminary multi-omic studies suggest that stress-response pathways and epigenetic reprogramming within oligodendroglial and neuronal lineages may contribute to MOGHE in the absence of coding mutations, indicating broader molecular heterogeneity [49].

## 7. Conclusions and Perspectives

Over the past decade, research on *SLC35A2*-related disorders has substantially advanced our understanding of how defective glycosylation impacts brain development and function. Both germline and somatic variants converge on impaired UDP-galactose transport, resulting in hypogalactosylation of N-glycans and downstream cellular, structural, and network-level abnormalities. These defects manifest as a spectrum of clinical phenotypes, from systemic *SLC35A2*-CDG with global neurodevelopmental impairment to focal cortical malformations such as MOGHE, which are associated with early-onset epilepsy and frequently amenable to surgical intervention. Histopathological analyses, patient-derived tissue studies, and experimental models have clarified that epileptogenesis in MOGHE may involve dysfunction in excitatory neurons, although the relative contribution of oligodendroglial abnormalities remains to be fully defined and could also play a role in shaping the epileptogenic substrate. Interestingly, a subset of histologically confirmed MOGHE cases do not show detectable *SLC35A2* variants, indicating that additional molecular or regulatory mechanisms may converge on the same pathological phenotype. Investigating these cases is essential to fully define the spectrum of MOGHE and to identify alternative pathogenic pathways that could inform both diagnosis and therapy.

Therapeutically, surgical resection remains the most effective approach for seizure control in MOGHE, while metabolic interventions such as oral D-galactose supplementation offer the potential to partially restore glycosylation and improve both biochemical and clinical outcomes in systemic and focal forms of *SLC35A2*-related disease. Emerging studies, including analyses of lesion-restricted somatic mosaicism and sex chromosome alterations, suggest additional layers of genetic and molecular complexity that could refine patient stratification and may contribute to mechanistic understanding, although these hypotheses remain preliminary and require validation in independent cohorts.

Future research will benefit from integrative approaches combining molecular genetics, single-cell and spatial multi-omic analyses, and refined in vivo and human iPSC-derived models (Table 1). Such strategies will be critical to delineate the full spectrum of *SLC35A2*-dependent cortical malformations, uncover cell-type-specific pathogenic mechanisms, and identify novel therapeutic targets. Ultimately, continued integration of mechanistic insights with precision diagnostics and interventions holds the promise of improving outcomes for patients across the broad spectrum of *SLC35A2*-related disorders, while also providing fundamental insights into glycosylation-dependent neurodevelopment and focal epileptogenesis.

## Figures and Tables

**Figure 1 ijms-26-11560-f001:**
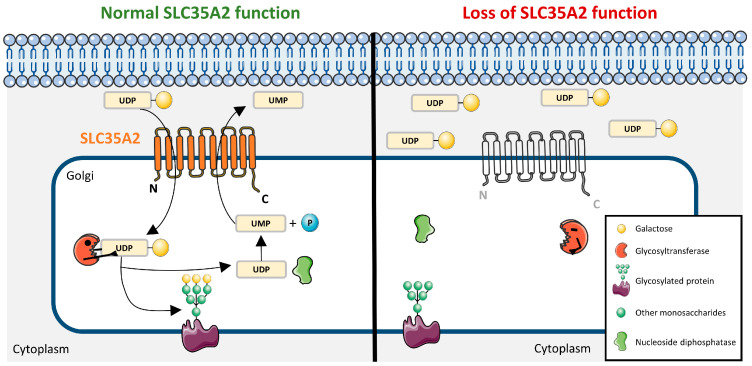
SLC35A2 function in glycosylation under normal and loss-of-function conditions. The schematic illustrates the role of the SLC35A2 transporter in UDP-galactose transport and subsequent glycosylation. In normal condition SLC35A2 (orange) facilitates the import of UDP-galactose from the cytoplasm into the Golgi apparatus. Within the Golgi, glycosyltransferases use UDP-galactose to add galactose residues to glycoproteins and other glycoconjugates. Nucleoside diphosphatase recycles UDP to UMP (uridine monophosphate), sustaining transport. In pathological conditions, loss of SLC35A2 reduces UDP-galactose import, leading to deficient glycosylation of proteins and accumulation of UDP-galactose in the cytoplasm.

**Figure 2 ijms-26-11560-f002:**
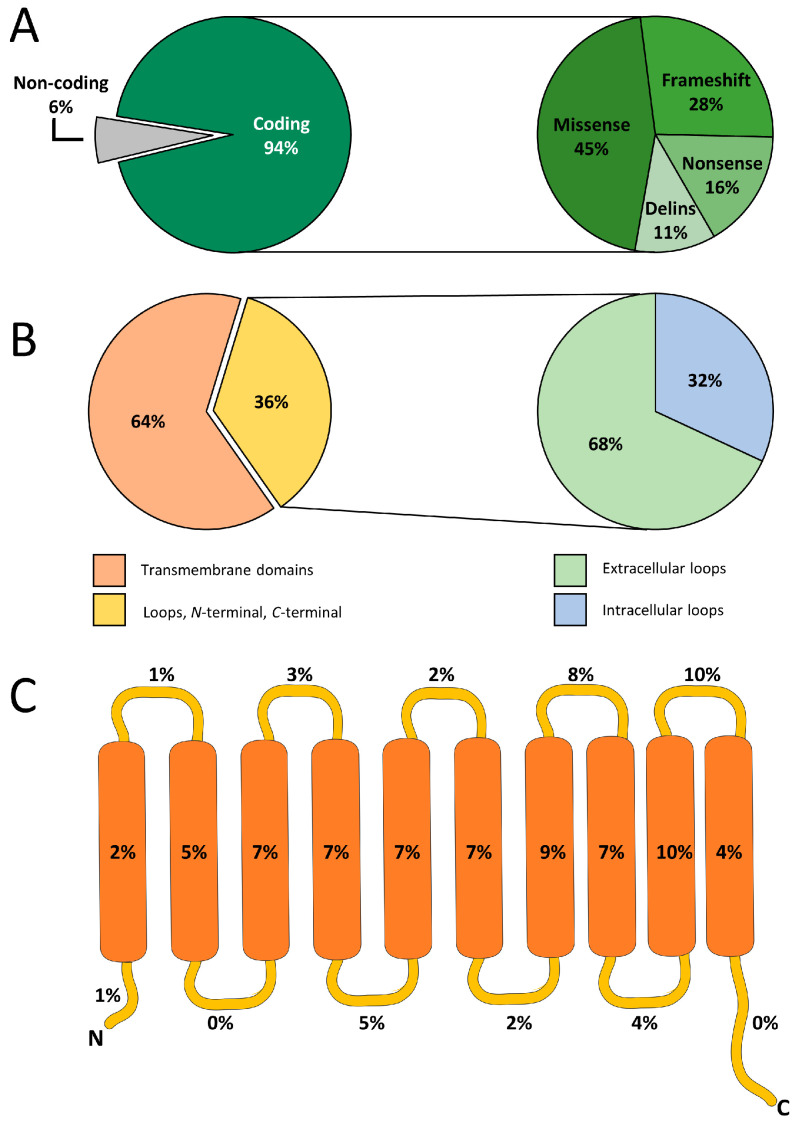
Distribution and localization of mutations in the human *SLC35A2* gene. (**A**) Pie charts showing the proportion of coding versus non-coding mutations (**left**) and the distribution of mutation types within coding regions (**right**). The vast majority (94%) of identified variants are located within coding regions, while 6% occur in non-coding regions. Among coding variants, missense mutations account for 45%, frameshift mutations for 28%, nonsense mutations for 16%, and small deletions/insertions (delins) for 11%. (**B**) Spatial distribution of *SLC35A2* mutations across protein domains. Most variants (64%) occur within transmembrane domains, while 36% are located in loops or terminal regions. Within the loop regions, 68% of mutations localize to intracellular loops and 32% to extracellular loops. (**C**) Schematic representation of the SLC35A2 protein topology, showing ten predicted transmembrane helices (orange) connected by intracellular and extracellular loops (yellow). Percentages indicate the relative frequency of mutations mapped to each region. The detailed dataset of the mutations used to generate these graphs is provided in Appendix A.

**Figure 3 ijms-26-11560-f003:**
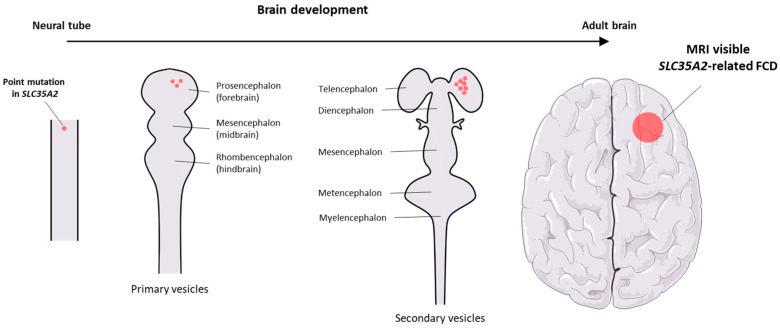
Somatic *SLC35A2* mutations leading to Focal Cortical Dysplasia (FCD). Simplified schematic illustrating how somatic mutations in *SLC35A2* can occur at any stage of brain development and result in FCD.

**Figure 4 ijms-26-11560-f004:**
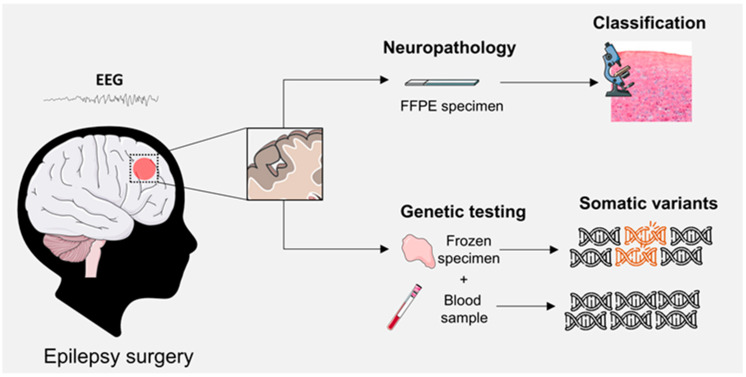
Workflow for the identification of *SLC35A2*-related cortical malformations at histopathological and genetic levels. The epileptogenic focus is defined through the correlation of MRI and EEG data, guiding epilepsy surgery. Resected tissue is examined by neuropathology for cortical lesion classification and by genetic testing of frozen brain and blood samples to detect somatic variants. Thus, integration of morphological and molecular findings enables a comprehensive diagnosis of focal cortical malformations. FFPE, Formalin-Fixed and Paraffin-Embedded.

**Figure 5 ijms-26-11560-f005:**
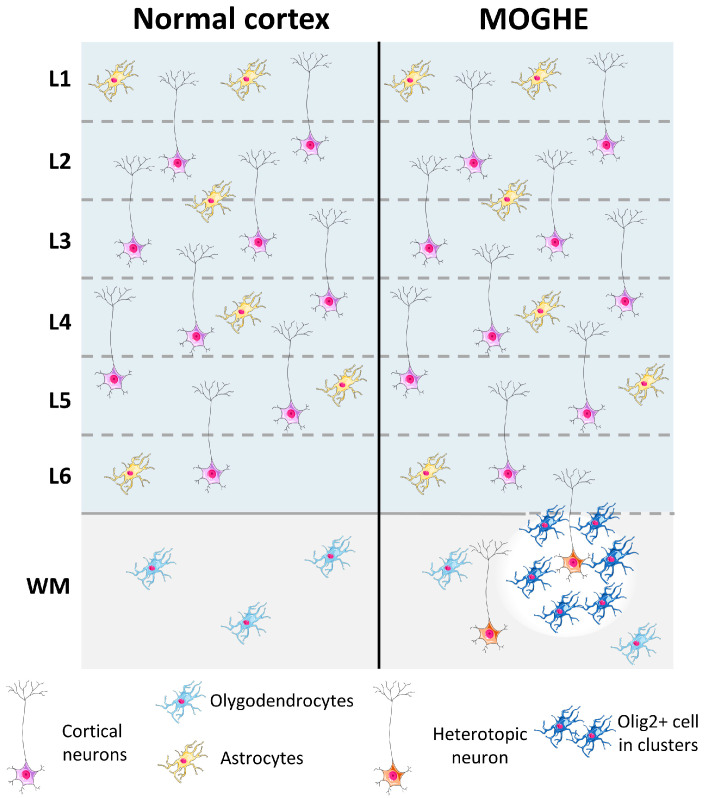
Neuropathological Features of MOGHE. The MOGHE phenotype is characterized by focal accumulations of oligodendrocyte-lineage cells within the white matter and deep cortical layers, a blurred grey–white matter boundary (dotted line), regions of irregular or reduced myelination (depicted by color gradients in the white matter), and the presence of ectopic neurons scattered throughout the white matter. L1–L6: layers 1–6; WM: White Matter.

**Table 1 ijms-26-11560-t001:** Future directions and open questions in *SLC35A2*-Related Disorders.

AREA	KEY OPEN QUESTIONS/RESEARCH PRIORITIES
MOLECULAR MECHANISMS	How does impaired UDP–galactose transport alter neuronal membrane composition, receptor trafficking, and network excitability? Integration of glycoproteomics and electrophysiology is needed to link hypogalactosylation to epileptogenesis.
GENETIC HETEROGENEITY	Which additional genes or regulatory mechanisms cause MOGHE in *SLC35A2*-negative cases? Deep sequencing and methylomic profiling should be applied to unresolved patients.
SEX CHROMOSOME BIOLOGY	What is the role of Y-chromosome mosaicism in MOGHE pathogenesis, and how does it interact with glycosylation and oligodendroglial proliferation?
CELLULAR AND TRANSLATIONAL MODELS	Develop neuronal–glial co-culture systems and brain organoids to replicate somatic mosaicism and dissect cell-type–specific dysfunction.
THERAPEUTIC STRATEGIES	Conduct controlled trials of D-galactose and explore complementary approaches (metabolic boosters, mRNA or viral rescue) to restore glycosylation.
CLINICAL INTEGRATION	Standardize diagnostic workflows combining MRI, EEG, and deep brain sequencing to improve identification and management of mosaic *SLC35A2*-related epilepsies.

## Data Availability

No new data were created or analyzed in this study. Data sharing is not applicable to this article.

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
