# Peer review of "SLC35A2-Related Brain Disorders: Genetics, Pathophysiology, and Therapeutic Insights"

_ijms, 2025, doi:10.3390/ijms262311560_

Round 1
Reviewer 1 Report
Comments and Suggestions for Authors The review presented by Risso et al. can be a valuable source of information about brain disorders (Congenital disorder of glycosylation, Epilepsy) caused by mutations in the SLC35A2 gene (member of the solute carrier (SLC) superfamily). The authors provide a detailed analysis of the molecular and cellular pathogenesis of SLC35A2-related disorders. In a separate section, they discuss the cases and prospects for successful treatment of these disorders with oral D-galactose supplementation. In my opinion, the topic was presented in an interesting and accessible way, the manuscript contains all the relevant information. The paper is structured; the 5 figures and 1 table complement the text. Abstract gives the necessary information about the contents of the paper. The reference list covers the relevant literature (the authors cite 57 references). After making several technical corrections (listed below), the review can be published in its current form.1. Section 2 has only one subsection—2.1.—which looks odd.
2. The legends for Figures 3 and 4 should better explain the abbreviations "FCD" and "FFPE," respectively. 3. If possible, it would be useful to include a section with a list of explanations of abbreviations, as many specific abbreviations are used in the review. 4. Two figures with number 4, please correct.
Author Response
1. Section 2 has only one subsection—2.1.—which looks odd.
Section 2.1 has been removed, and the content has been integrated directly into Section 2 to avoid having a single subsection.
2. The legends for Figures 3 and 4 should better explain the abbreviations "FCD" and "FFPE," respectively
Figure legends for Figures 3 and 4 have been revised to clearly explain the abbreviations “FCD” (Focal Cortical Dysplasia) and “FFPE” (Formalin-Fixed, Paraffin-Embedded).
3. 3. If possible, it would be useful to include a section with a list of explanations of abbreviations, as many specific abbreviations are used in the review.
A comprehensive list of abbreviations has been added to facilitate readability, given the large number of specialized terms used throughout the review.
4. Two figures with number 4, please correct.
The numbering conflict with two Figures labeled as 4 has been corrected.
Reviewer 2 Report
Comments and Suggestions for Authors
The manuscript “ SLC35A2-Related Brain Disorders: Genetics, Pathophysiology, and Therapeutic Insights” by Antonio Falace et al. is a comprehensive Review of the title disorders. It is well referenced, and does not require substantial restructuring and synthesis to meet publication standards. Having in mind that prof. A. Falace is the guest editor of the special issue the manuscript probably aims at being introductory manuscript. Thus some minor adjustments are needed.
The review, lists studies sequentially, without sufficient synthesis, critical comparison, or identification of controversies. Each major section may conclude with a concise, analytical summary e.g. : While the core framework of SLC35 family function is established, unresolved questions remain: NST complex formation, oligomerization, regulation by metabolic pathways, or the mechanistic basis of tissue-specific disease phenotypes. Addressing these gaps … On some sections such conclusions are present.
Figure numbering is inconsistent, with two figures labeled “Figure 4”.
Certain statements overinterpret preliminary findings, particularly regarding:
neuron-centric vs. oligodendroglial contributions, the mechanistic role of Y-chromosome mosaicism (preprint).
The manuscript may include problems such as sampling bias in resected brain tissue,
variation in detection sensitivity among centers, a little bit more ideas on unresolved molecular causes in SLC35A2-negative MOGHE.
Sections 3.2, 3.3, and 4 contain very long paragraphs that may be “split”
The discussion of D-galactose supplementation is too narrow/slim. Additional perspectives (gene therapy, mRNA , ...) may be discussed.
Minor
L194 et al >> et al.
L202 excitatory–inhibitory balance
Author Response
1.The review, lists studies sequentially, without sufficient synthesis, critical comparison, or identification of controversies. Each major section may conclude with a concise, analytical summary e.g. : While the core framework of SLC35 family function is established, unresolved questions remain: NST complex formation, oligomerization, regulation by metabolic pathways, or the mechanistic basis of tissue-specific disease phenotypes. Addressing these gaps … On some sections such conclusions are present
We thank the reviewer for this insightful suggestion. In the revised manuscript, we have incorporated concise, analytical summaries at the end of each major section wherever possible.
Figure numbering is inconsistent, with two figures labeled “Figure 4”
This has been corrected
Certain statements overinterpret preliminary findings, particularly regarding:
neuron-centric vs. oligodendroglial contributions, the mechanistic role of Y-chromosome mosaicism (preprint).
We thank the reviewer for highlighting this point. We have carefully revised the manuscript to ensure that preliminary findings are not overinterpreted. Specifically, the discussion regarding neuron-centric versus oligodendroglial contributions and the potential mechanistic role of Y-chromosome mosaicism (preprint) has been tempered and clarified. These considerations have been explicitly addressed in Section 3.2 and in the Conclusions, highlighting their preliminary nature and the need for further validation.
The manuscript may include problems such as sampling bias in resected brain tissue,
variation in detection sensitivity among centers, a little bit more ideas on unresolved molecular causes in SLC35A2-negative MOGHE.
We thank the reviewer for this constructive comment. In the revised manuscript, we have added a new section, “Limitations and Additional Considerations” (Section 6), which addresses issues including sampling bias in resected brain tissue, variation in detection sensitivity among centers, and potential unresolved molecular causes in SLC35A2-negative MOGHE. This section highlights current gaps and areas for future investigation.
Sections 3.2, 3.3, and 4 contain very long paragraphs that may be “split”
We thank the reviewer for this suggestion. In the revised manuscript, long paragraphs in Sections 3.2, 3.3, and 4 have been split into shorter, more readable paragraphs, improving clarity and readability without altering the content.
The discussion of D-galactose supplementation is too narrow/slim. Additional perspectives (gene therapy, mRNA , ...) may be discussed.
We thank the reviewer for this comment. In the revised manuscript, the discussion on therapeutic perspectives has been expanded at the end of the relevant section to include additional approaches beyond D-galactose supplementation, such as gene therapy, mRNA-based strategies, and other emerging precision medicine interventions.
Minor
L194 et al >> et al.
L202 excitatory–inhibitory balance
These have been corrected